# The Effect of Parents’ Nutrition Literacy on Children’s Oral-Health-Related Quality of Life

**DOI:** 10.3390/ijerph21091190

**Published:** 2024-09-08

**Authors:** Busra Aslan Gonul, Betul Cicek

**Affiliations:** 1Department of Nutrition and Dietetics, Faculty of Health Sciences, Institute of Health Sciences, Erciyes University, 38280 Kayseri, Turkey; 2Department of Nutrition and Dietetics, Faculty of Health Sciences, Erciyes University, 38280 Kayseri, Turkey; bcicek@erciyes.edu.tr

**Keywords:** nutrition literacy, oral-health-related quality of life, child oral health

## Abstract

Parents can help children adopt good eating habits early in childhood by encouraging them to eat healthy. While parents’ levels of nutritional literacy is known to play a role in children’s healthy nutrition, healthy food is also essential in improving oral-health-related quality of life (OHRQoL). Since the effect of parents’ nutritional literacy on children’s OHRQoL is not yet known, this study aimed to examine the impact of parental nutritional literacy on children’s OHRQoL. This study was conducted with 459 parents of children aged 3–6 living in the Central Anatolia Region of Türkiye. Data collection tools included a sociodemographic form, the Early Childhood Oral Health Impact Scale (ECOHIS) and the Evaluation Instrument of Nutrition Literacy on Adults (EINLA). Data were analyzed in SPSS, using Student’s *t*-test, Chi-square test, and binary logistic regression models. Parents with higher nutritional literacy tend to have higher levels of education. An increase in total nutritional literacy score, especially in the general nutrition knowledge (GNK) and food label and numerical literacy (FLNL) subscales, was associated with higher child OHRQoL. Increasing parents’ nutritional literacy levels can contribute to OHRQoL by enabling them to be good role models for their children. Therefore, increasing parents’ nutritional literacy can improve OHRQoL by improving children’s nutrition and can be considered a protective factor for oral health.

## 1. Introduction

The period between ages 3 and 6 is critical for children’s physical and mental development and eating habits [1,2]. Children’s eating habits are influenced by several factors, including family environment, dietary style, and social environment [2]. During this period, parents can create a healthy environment at home and be role models for children, which can help them adopt good eating habits at a young age [1]. Practices such as sitting at the table with children and encouraging them to eat healthy snacks positively affect children’s eating behavior [3]. Since parents decide what their children will eat and when they will eat it, their nutritional literacy level significantly impacts children’s eating habits [4,5].

Nutrition literacy may be defined as an individual’s level of acquiring, reading, and understanding basic nutritional information. Nutrition literacy is also defined as the ability to read and understand complex food information and provides an understanding of the primary food groups (carbohydrates, proteins, and fats), food sources, and their roles in maintaining health [6]. Studies have shown that parents’ nutritional literacy levels are closely related to their children’s nutrition [1,7]. Parents with adequate nutritional literacy score higher on applying healthy nutrition guidance and monitoring children’s unhealthy food consumption [1]. Similarly, mothers with a high level of nutritional literacy feed their children more vegetables, fruits, legumes, low-sugar drinks, and foods with less artificial content than mothers with a low level of nutritional literacy [8]. In another study, every 1% increase in parental nutritional literacy was associated with a 0.51-point increase in children’s diet quality [7]. Improving children’s diet quality and healthy eating behaviors is closely associated with increased health-related quality of life [9].

Health-related quality of life is a broad concept closely related to physical, mental, and social health [10]. Oral-health-related quality of life (OHRQoL), a subcomponent of health-related quality of life, expresses the positive or negative impact of oral-health-related factors on the individual’s quality of life [11]. OHRQoL, especially in early childhood, may affect children’s growth and development, socialization, and daily activities [10]. Parental socioeconomic status, education level, oral health behaviors, and nutritional factors affect the children’s OHRQoL [11]. It is important for parents to provide oral health care to their children from an early age and help them brush their teeth [12]. It has been shown that oral health practices such as the age at which children start brushing their teeth from infancy, regular tooth brushing, a habit of visiting the dentist [13], and proper nutrition practices at home [14] contribute to oral health by significantly reducing children’s need for dental treatment. Since children often learn and implement their parents’ behaviors, oral health practices and habits of parents and thus their children may influence children’s OHRQoL [11]. Therefore, oral health protective practices such as healthy nutrition are essential in improving OHRQOL [15].

Parental nutritional behaviors such as parent–child nutrition education, consumption of healthy snacks and drinks at home, limiting the consumption of junk food and sugary drinks and milk, and not giving unhealthy foods to children to regulate their mood are important points that affect children’s oral health [12]. It has been stated that consuming two or more snacks per day at home adversely affects children’s oral health, and therefore it is important for parents to be controlled and conscious of their feeding practices [14]. Studies have found that nutritional factors such as natural milk consumption and time spent eating cariogenic foods are significantly associated with OHRQoL [11,16]. As can be seen from this literature review, children’s OHRQoL and nutrition are closely interrelated [11,16]. For this reason, we think that the nutritional literacy levels of the parents, especially in early childhood between 3 and 6 years of age, may affect children’s OHRQoL. However, to our knowledge, no studies focus on the impact of parental nutritional literacy on children’s OHRQoL in early childhood. This research is important because it addresses a clear gap in the literature and focuses on this understudied area of the impact of parental nutritional literacy on children’s OHRQoL. This study aims to provide valuable information for interventions by highlighting the importance of parental nutritional literacy in children’s OHRQoL. Therefore, the main objective of this study was to determine the effect of nutritional literacy of parents of children aged 3–6 years on children’s OHRQoL.

The objectives are:Determining the OHRQoL of children aged 3–6 by asking their parents;Determining the nutritional literacy of parents;Determining the relationship between nutritional literacy of parents and children’s OHRQoL.

## 2. Materials and Methods

### 2.1. Participants and Procedure

This cross-sectional and descriptive study was performed with parents of children aged 3–6 in a city in the Central Anatolia Region of Türkiye in 2023. Parents who consented to participate in the study could understand and answer the questions and observe the child’s daily food consumption. Parents whose children had a food allergy or an acute or chronic disease that would affect food intake or whose children needed to follow a specific diet related to the disease were excluded. The sample of this cross-sectional study is at least 384 people with a 5% margin of error at a 95% confidence level calculated by the sample size with an unknown population.

### 2.2. Ethical Aspect of Research

Our study was conducted following the ethical rules and the principles of the Declaration of Helsinki. All participants were informed about the purpose of the study, and their written informed consent was obtained face to face. Ethical approval was obtained from the Erciyes University Social and Humanities Science Ethics Committee (Date: 29 November 2022, Approval number: 504). The data were collected from voluntary participants via a face-to-face survey form. The participants were given the necessary information about the research, and were told that the information they gave would remain confidential and would not be used anywhere other than in the research to make them feel comfortable. Informed consent was obtained from all individual participants included in the study.

### 2.3. Measures

A questionnaire including a personal information form requesting information on the parent and child, the Early Childhood Oral Health Impact Scale (ECOHIS), and the Adult Nutrition Literacy Assessment Tool (EINLA) was administered to the parents. In the personal information form, general information, health information, oral health practices, anthropometric measurements, and nutritional habits of parents and children were questioned. Children’s tooth decay was determined according to the families’ self-reports. Parental body mass index (BMI) values were calculated by dividing their weight in kilograms by height in square meters, and the World Health Organization’s classification for adults was used in the BMI assessment of the parents [17]. For children, weight-for-age and BMI-for-age growth curves developed by the World Health Organization were used [18].

### 2.4. Early Childhood Oral Health Impact Scale (ECOHIS)

ECOHIS was developed by Pahel et al. [19] to assess the influence of oral problems and treatments on quality of life in preschool children. This study was used to measure children’s OHRQOL ECOHIS and its Turkish validity and reliability, which was tested by Peker et al. [17]. The scale questions were designed for the parents of the children, considering that they better relate the effects of oral health to the quality of life since children in this age group do not have sufficient psychosocial and developmental maturity to do so. It is based on answering 13 questions in two main sections: child and family impact subsections. The child impact section includes symptoms, function, psychology, self-image, and social interaction. The family impact section comprises parental distress and family function. The answers to each question were categorized as never (0 points), rarely (1 point), sometimes (2 points), often (3 points), very often (4 points), and unknown (5 points). Higher scores signify a more significant oral health impact and lower OHRQoL [20].

### 2.5. Evaluation Instrument of Nutrition Literacy on Adults (EINLA)

This questionnaire, which consists of five subscales, was created by Cesur et al. [21] and was used to measure nutrition literacy. These subscales are for general nutrition knowledge (GNK), reading comprehension and interpretation (RCI), food groups (FG), portion sizes (PS), and food labels and numerical literacy (FLNL), respectively. Responses to the items are assigned scores of either 1 (correct) or 0 (incorrect/I do not know), with attainable scores ranging from a minimum of 0 to a maximum of 35. The scale is divided into three levels: a score of 24–35 points is categorized as sufficient, 12–23 points as borderline, and 0–11 points as insufficient [21].

### 2.6. Data Analysis

SPSS version 27.0 software was used for statistical analysis, and a *p*-value < 0.05 was considered significant. Number (n) and percentage (%) were calculated for categorical variables, and arithmetic mean and standard deviation were calculated for continuous variables. The conformity of the scale scores to normality was evaluated using the Kolmogorov–Smirnov test, the histogram, and normal Q–Q graphs. The EINLA scores categorized parental and child characteristics, anthropometric measurements, and all scale scores. Chi-square tests for categorical variables and Student *t*-tests for parametric variables were used to evaluate characteristic differences between EINLA results. Binary logistic regression analysis determined the relationship between EINLA (independent variables) and ECOHIS (dependent variables). In this analysis, the dependent variable is ECOHIS, the independent variable is EINLA, and the covariate variables are the child’s age, gender, BMI z-score, consumption of acidic beverages, brushing habits, and decay, and parental age, relationship to the child, BMI, education status, family income, brushing habits, and frequency of visiting the dentist. The borderline nutrition literacy category of EINLA and its subscales was considered the reference group. Borderline and sufficient nutrition in parents were compared to estimate the risk of OHRQoL in children. To perform regression analyses, we first evaluated possible confounding variables one by one using univariate analysis. For multivariate analysis, we selected variables for which the Wald test was significant at the 0.25 level [22]. Next, we assessed each potential confounding variable in the multivariate models. We retained it as a confounder if it substantially increased Nagelkerke or adjusted R2 values and/or changed the relevant association. Model 1 was adjusted for the child’s age and gender. In Model 2, in addition to the child’s age and gender, the child’s BMI z-score, consumption of acidic beverages, brushing habits, and decay and the parental age, relationship to the child, BMI, education status, family income, brushing habits, and frequency of visiting the dentist were adjusted.

## 3. Results

The sample consisted of 459 parent–child pairs. The mean age of the parents was 34.1 ± 5.5 years, and 76.5% were mothers. Parents’ nutritional literacy score was 28.5 ± 3.4, and mothers’ scores were higher than fathers’ (*p* = 0.035). Parents in the sufficient nutrition literacy category had higher education status than those in the borderline category (*p* < 0.05). Parents in the sufficient nutrition literacy category brushed their teeth more regularly than those in the borderline category (*p* < 0.05). The mean age of the children was 56.3 ± 12.2 months, and 46.8% were girls. The rate of not brushing teeth was lower in children whose parents had sufficient nutritional literacy (*p* < 0.05). Children of parents with sufficient nutritional literacy had better OHRQoL than those with borderline nutritional literacy (*p* < 0.05). Sociodemographic characteristics grouped according to nutritional literacy are shown in Table 1.

Odds ratios (OR) and 95% CI for child OHRQoL by parental nutrition literacy scores are provided in Table 2. Binary logistic regression analyses were performed to determine the effect of high nutrition literacy on high OHRQoL. In the crude model, high total nutrition literacy score (OR 2.66, 95% CI 1.25–5.66; *p* = 0.011) and subscales of GNK (OR 1.94, 95% CI 1.24–3.06; *p* = 0.04), PS (OR 1.81, 95% CI 1.20–2.99; *p* = 0.02), and FLNL (OR 2.07, 95% CI 1.38–3.09; *p* < 0.001) were associated with high OHRQoL of children. In Model 1, the child’s age and gender are included as confounding variables. In Model 2, we adjusted for the child’s age, gender, BMI z-score, consumption of acidic beverages, brushing habits, and decay, and parental age, relationship to the child, BMI, education status, family income, brushing habits, and frequency of dentist visits. Higher total parental nutrition literacy scores were associated with an elevated child OHRQoL (OR 4.01, 95% CI 1.60–10.50; *p* = 0.003). That is, for a total EINLA score > 24, the odds of developing child OHRQoL are greater by a factor of 4.01. This association was also detected for the GNK (OR 2.37, 95% CI 1.38–4.08; *p* = 0.002) and FLNL (OR 2.47, 95% CI 1.52–4.00; *p* < 0.001) subscales. The PS subscale was associated with increased OHRQoL in Model 1 (OR 1.78, 95% CI 1.07–2.95; *p* = 0.03); however, it was not in Model 2.

## 4. Discussion

This study was conducted to determine the effect of the nutritional literacy of parents of children aged 3–6 on the children’s OHRQoL. For the first time in the literature, we found that an increase in parents’ nutritional literacy scores was associated with an increase in children’s OHRQoL. The increase in parents’ total nutritional literacy scores, mainly with subscales such as general dietary knowledge, food label literacy, and numerical literacy, positively affected children’s OHRQoL. Since it is a subcomponent of children’s health-related quality of life, determining OHRQoL and the factors affecting it is especially important in connection with children’s physical, mental, and social health. Considering the importance of these issues, it is thought that the current findings will shed light on the literature and lead to clinical studies.

Nutritional literacy is fundamental for families to access accurate nutrition information, analyze it, make the right food choices, maintain healthy eating behavior, and apply these correctly to their children [5]. As parents’ level of education increases, their nutritional literacy also increases [5,6,7]. Studies have shown that parents with adequate nutritional literacy have significantly higher levels of education than others and that increases in nutritional literacy are associated with increases in educational levels [5,6]. In the current study, similar to the existing literature, the parents’ education level in the group with high nutritional literacy was higher. This situation can be explained by the fact that parents with elevated levels of education care more about health, are more conscious about nutrition, and have greater access to information about it. Similarly, it is thought that the high rate of regular tooth brushing in parents with high nutritional literacy and their children is due to their health awareness. Accordingly, it is essential to draw attention to the role of the education level of families in nutritional literacy and the potential positive effect of this factor on the oral hygiene of both them and their children.

The concept of OHRQoL relates to the impact of oral health or disease on a person’s well-being or quality of life [23]. OHRQOL, a vital subcomponent of general health-related quality of life, has been accepted by the World Health Organization as an essential part of the Global Oral Health Program [24]. Adopting preventive oral health behaviors such as healthy nutrition is an economical and indispensable way to improve OHRQOL [15]. Incorrect nutritional practices, especially in relation to the age at which sugar is given and the frequency of consumption, can negatively affect oral health and the child’s OHRQoL in early childhood [25]. Children in early childhood develop healthy eating patterns by learning their parents’ eating habits and dietary practices. It has been shown that children not only follow their parents’ instructions but also imitate their parents’ behavior in terms of nutrition [26]. It has been shown that parents with high nutritional literacy create a healthier nutritional environment for their children and do not use foods as rewards for emotion regulation [1]. Therefore, parents must update their nutritional literacy and create a healthy eating environment for their children to improve QHRoL in early childhood. Although studies often refer to the relationship between parents’ oral health literacy and child OHRQoL, or health literacy and child oral health [27,28,29], this relationship with nutritional literacy has not yet been examined. Current findings have shown that children of parents with sufficient nutritional literacy have better OHRQoL than children of parents in the borderline category. The parents’ average EINLA scale score of 28.5 ± 3.4 indicates their sufficient nutritional literacy. Similar to our results, in studies conducted with mothers of children aged 3–6, the average nutritional literacy scores were 27.22 ± 3.45 and 27.9 ± 4.64, respectively [5,6]. For a total parental EINLA score > 24, the odds of the child developing OHRQoL are greater by 4.01 times. This relationship was also detected for the GNK (OR 2.37, %95 CI 1.38–4.08; *p* = 0.002) and FLNL (OR 2.47, %95 CI 1.52–4.00; *p* < 0.001) subscales. The PS subscale was associated with increased child OHRQoL in Model 1 (OR 1.78, 95% CI 1.07–2.95; *p* = 0.03), but not, however, in Model 2.

### Limitations and Future Research

The current study has several limitations. The study limitations include the cross-sectional design, which prevents causal inference. In addition, questioning the nutritional literacy of both parents could have provided a more comprehensive approach. All participants’ anthropometric values were based on self-reporting, and parents living in a specific region of Türkiye were included in the study. Since it assessed OHRQoL, the contribution of a dentist to the study could have made it more valuable. However, since there is no literature evaluating the effect of parental nutritional literacy on child OHRQoL, the results of this study suggest that high nutritional literacy in parents can improve child OHRQoL. In studies, children’s BMI z scores [1] and snack consumption such as of soft drinks [3], and parental age [4], BMI [1], education level [4], and income status [4] were associated with parental nutritional literacy, while children’s food consumption [25] and children’s and parents’ oral health behaviors and practices [10,11] were associated with child OHRQoL. In addition, while some studies have included both parents, others included only mothers, which led us to consider the relationship to the child as a confounding factor [4,8]. Therefore, these variables were considered important confounders for our study and adjustment was made for these variables. Our results can also form the basis for larger-sample clinical studies to improve parental nutritional literacy and determine the effects of parental nutritional literacy on child OHRQoL.

## 5. Conclusions

This study has shown for the first time that parental nutritional literacy affects children’s OHRQoL. Determining the factors that affect OHRQoL in early childhood is especially important for designing effective interventions. Therefore, increasing parents’ nutritional literacy levels will improve children’s OHRQoL and oral health by providing suitable role models. Training can be provided in future studies to increase parents’ nutritional literacy, and the effects on children’s OHRQoL can be determined.

## Figures and Tables

**Table 1 ijerph-21-01190-t001:** General characteristics of parents and children according to nutrition literacy classification.

Variables ^a^			All Participants (n = 459)	Nutrition Literacy (n = 459)		
				Borderline Nutrition Literacy	Sufficient Nutrition Literacy	*p*
Parent’s variables	Age (years)		34.1 ± 5.5	34.5 ± 6.0	34.1 ± 5.4	0.654
	BMI (kg/m^2^)		25.5 ± 4.3	25.9 ± 5.2	25.5 ± 4.2	0.575
	Relationship to the child	Mother	351 (76.5)	28 (77.8)	323 (76.4)	0.847
		Father	108 (23.5)	8 (22.2)	100 (23.6)	
	Education status	College or below	197 (42.9)	29 (80.6)	168 (39.7)	<0.001 †
		University or above	262 (57.1)	7 (19.4)	255 (60.3)	
	Family income	Low	100 (21.8)	9 (25.0)	91 (21.5)	0.350
		Medium	250 (54.5)	22 (61.1)	228 (53.9)	
		High	109 (23.7)	5 (13.9)	104 (24.6)	
	Brushing teeth	Brushing regularly	319 (69.5)	17 (47.2)	302 (71.4)	0.003 †
		Brushing irregularly	133 (29.0)	17 (47.2)	116 (27.4)	
		Not brushing	7 (1.5)	2 (5.6)	5 (1.2)	
	Frequency of dentist visits	Regularly	53 (11.5)	5 (13.9)	48 (11.3)	0.309
		Sometimes	91 (19.8)	9 (25.0)	82 (19.4)	
		Only times of pain	268 (58.4)	16 (44.4)	252 (59.6)	
		Never	47 (10.2)	6 (16.7)	41 (9.7)	
Child’s variables	Age (months)		56.3 ± 12.2	54.0 ± 11.1	56.5 ± 12.3	0.246
	Gender (girl)		215 (46.8)	14 (38.9)	201 (47.5)	0.319
	BMI-for-age z scores		0.35 ± 1.7	−0.05 ± 1.82	0.38 ± 1.65	0.127
	Decay (Yes)		173 (37.7)	16 (44.4)	157 (37.1)	0.384
	Brushing teeth	Brushing regularly	192 (41.8)	14 (38.9)	178 (42.1)	0.021 †
		Brushing irregularly	207 (45.1)	12 (33.3)	195 (46.1)	
		Not brushing	60 (13.1)	10 (27.8)	50 (11.8)	
	Consumption of acidic beverages	Always	105 (22.9)	11 (30.6)	94 (22.2)	0.520
		Sometimes	326 (71.0)	23 (63.9)	303 (71.6)	
		Never	28 (6.1)	2 (5.6)	26 (6.1)	
	ECOHIS score		5.1 ± 7.5	8.9 ± 10.3	4.8 ± 7.1	0.027 *

BMI: Body mass index. * Student’s *t*-test, *p* <0.05. † Chi square test, *p* <0.05. ^a^ Values were expressed as the number (n) and percentage (%) for qualitative variables and mean ± standard deviation for quantitative variables.

**Table 2 ijerph-21-01190-t002:** Logistic regression analysis assessing the associations between EINLA and OHRQoL.

Variables ^a^	OHRQoL					
	Crude ^b^		Model 1 ^c^		Model 2 ^d^	
	OR	95% CI	OR	95% CI	OR	95% CI
Total Nutrition Literacy Scores	2.66	1.25, 5.66 *	2.98	1.38, 6.41 *	4.01	1.60, 10.50 *
GNK	1.95	1.24, 3.06 *	2.04	1.29, 3.22 *	2.37	1.38, 4.08 *
RCI	1.32	0.81, 2.16	1.37	0.83, 2.25	1.78	0.96, 3.30
FG	1.48	0.51, 4.13	1.45	0.50, 4.21	2.29	0.64, 8.19
PS	1.81	1.10, 2.99 *	1.78	1.07, 2.95 *	1.59	0.89, 2.85
FLNL	2.07	1.38, 3.09 *	2.07	1.38, 3.12 *	2.47	1.52, 4.00 *

FG, food groups; FLNL, food label and numerical literacy; GNK, general nutrition knowledge; PS, portion sizes; RCI, reading comprehension and interpretation; OHRQoL, oral-health-related quality of life. * *p*-trend < 0.05. ^a^ Values are odds ratio (95% confidence interval) estimated through logistic regression using the low category of nutritional literacy and its subscales as reference. ^b^ Crude: not adjusted for any variables. ^c^ Model 1: the model was adjusted for the child’s age and gender. ^d^ Model 2: the model was adjusted for the child’s age, gender, BMI z-score, consumption of acidic beverages, brushing habits, and decay and parental age, relationship to the child, BMI, education status, family income, brushing habits, and frequency of visiting the dentist.

## Data Availability

All data generated or analyzed during this study are included in this article. Further enquiries can be directed to the corresponding author.

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
