# Peer review of "The Effect of Parents’ Nutrition Literacy on Children’s Oral-Health-Related Quality of Life"

_ijerph, 2024, doi:10.3390/ijerph21091190_

Round 1
Reviewer 1 Report
Comments and Suggestions for Authors
Thank you for allowing to review your manuscript. Overall, its a good and important study, however, I have following comments:
1- It would be good to include more literature in background section about how parents Nutrional and dental health literacy affect children's oral health
2- Clearly mention current manuscripts aims and objectives
3- Clearly mention existing gap in literature and therefore need of present study
4- In methods section clearly establish and describe dependent, independent, covariate variables of present study
5- In results section table two mention's model 1 and model 2, however, there is no mention about what variables those model included in the methods, or data analysis section of the manuscript. Please consider mentioning them.
6- In limitations and future research section please consider mentioning significant confounders from the study context perspective by citing relevant studies, and how it affects existing obtained estimates of regression analysis.
Comments on the Quality of English Language
Minor syntax error
Author Response
Comments 1: It would be good to include more literature in background section about how parents Nutrional and dental health literacy affect children's oral health.
Response 1: Thank you for your valuable contribution. Thanks to your guidance, more literature has been added to the introduction section on how parental nutrition and oral health habits affect children's oral health. Since oral health literacy is not addressed in the study, no literature has been added on this subject. You can view the literature additions on page 2, lines 57-64 and 66-71.
Comments 2: Clearly mention current manuscripts aims and objectives.
Response 2: Thank you pointing this out. We agree with this comment. We have, accordingly, revised aims and objectives to emphasize this point. We revised the purpose of the study and added objectives. You can view the additions made on page 2, line 80-89.
Comments 3: Clearly mention existing gap in literature and therefore need of present study.
Response 3: Thank you pointing this out. We agree with this comment. The existing gap in the literature and the importance of this study are highlighted. You can view the additions made on page 2, line 78-88.
Comments 4: In methods section clearly establish and describe dependent, independent, covariate variables of present study.
Response 4: Agree. We have, accordingly dependent, independent and covariate variables are clearly stated. You can view the additions made on page 4, line 155-158.
Comments 5: In results section table two mention's model 1 and model 2, however, there is no mention about what variables those model included in the methods, or data analysis section of the manuscript. Please consider mentioning them.
Response 5: Thank you pointing this out. We agree with this comment. Which variables were included in model 1 and model 2 were added to the data analysis section. You can view the additions made on page 4, line 165-169.
Comments 6: In limitations and future research section please consider mentioning significant confounders from the study context perspective by citing relevant studies, and how it affects existing obtained estimates of regression analysis.
Response 6: Thank you pointing this out. We agree with this comment. Important confounders are explained and mentioned with reference to relevant studies in the Limitations and Future Research section. You can view the additions made on page 7, line 277-284.
Reviewer 2 Report
Comments and Suggestions for Authors
Thank you for your interesting work. Can you please clarify the following.
1- was the questioner constructed first in English and the translated to Turkish, please provide details on how was this validated.
2- was the questioner tested and validated for proper apprehension before being administered, please clarify.
3- Minor English is requested.
Thank you
Comments on the Quality of English Language
Minor English editing
Author Response
Comments 1: Was the questioner constructed first in English and the translated to Turkish, please provide details on how was this validated.
Response 1: Thank you pointing this out. The Early Childhood Oral Health Impact Scale, one of the scales used, was prepared in Turkish and administered in Turkish because its validity and reliability were done and translated into Turkish. Similarly, the Evaluation Instrument of Nutrition Literacy on Adults scale was also developed in Turkish and administered in Turkish. Other personal questions in the survey were also administered in Turkish and no translation was made.
Comments 2: Was the questioner tested and validated for proper apprehension before being administered, please clarify.
Response 2: Thank you pointing this out. Yes, the researchers first applied the survey among themselves and collected data after gaining experience. On the other hand, since the people who applied the survey were experts in their field (Nutrition and Dietetics department), this adaptation was quite fast.
Comments 3: Minor English is requested.
Response 3: Thank you very much for your valuable contribution. For the copyediting and proofreading service of this manuscript, language support and a certificate were received from Proofreading & Editing Office of the Dean for Research at Erciyes University. I am attaching this certificate as a file. I hope it will be sufficient for you.
